# Addressing Mistake Severity in Neural Networks with Semantic Knowledge

**Natalie Abreu**
MIT Lincoln Laboratory
Lexington, MA 02421
Natalie.Abreu@ll.mit.edu

**Nathan Vaska**
MIT Lincoln Laboratory
Lexington, MA 02421
Nathan.Vaska@ll.mit.edu

**Victoria Helus**[*]
MIT Lincoln Laboratory
Lexington, MA 02421
Victoria.Helus@ll.mit.edu

## Abstract

Robustness in deep neural networks and machine learning algorithms in general is an open research challenge. In particular, it is difficult to ensure algorithmic performance is maintained on out-of-distribution inputs or anomalous instances that cannot be anticipated at training time. Embodied agents will be deployed in these conditions, and are likely to make incorrect predictions. An agent will be viewed as untrustworthy unless it can maintain its performance in dynamic environments. Most robust training techniques aim to improve model accuracy on perturbed inputs; as an alternate form of robustness, we aim to reduce the severity of mistakes made by neural networks in challenging conditions. We leverage current adversarial training methods to generate targeted adversarial attacks during the training process in order to increase the semantic similarity between a model's predictions and true labels of misclassified instances. Results demonstrate that our approach performs better with respect to mistake severity compared to standard and adversarially trained models. We also find an intriguing role that non-robust features play with regards to semantic similarity.

## 1 Introduction

Traditionally, the success of a machine learning system is measured using straightforward metrics which treat all errors at the class level equally. However, this limited definition contradicts the natural intuition that some errors are far worse than others. As machine learning proliferates in embodied agents, distribution shifts, anomalies, and unique situations will make it extremely difficult to guarantee that all agents will operate without committing mistakes. Similarly, current definitions of robustness measure how well a machine learning system can maintain classification accuracy in a changing or degrading environment. We consider robustness from a different perspective: evaluating models not only on their accuracy, but also on the magnitude of the errors they commit. Using these metrics as additional objectives motivates the development of training techniques which prioritize lower error magnitudes.

Consider an autonomous home care robot that relies primarily on deep learning-based computer vision systems for object detection. As the robotic unit is deployed at scale and as the time since its deployment increases, the object detection system will be challenged by both environmental and

---

[*]corresponding author

36th Conference on Neural Information Processing Systems (NeurIPS 2022).

temporal distribution shifts leading to decreased detection accuracy. Moreover, a robot in these environments may end up committing costly mistakes; for example, it could misidentify a phone as a plate leading to the phone's destruction in a dishwasher or serve a houseplant for dinner instead of a salad. It can be assumed that a robotic product which commits such severe mistakes will not earn the trust of its user, and may not be retained. However, if the mistakes committed by the robot are minor they may be overlooked; an user is much more likely to forgive a robot for putting a place mat in the dishwasher instead of a phone.

When measuring mistake severity, higher penalties can be manually assigned to certain mistakes. However, this approach requires direct human input, and becomes unwieldy as the space of possible mistakes grows. As a stand-in for human-assigned penalties, we measure mistake severity as a function of the semantic similarity between the true and predicted classes. The motivation for this measure is intuitive; for example, consider an autonomous car which misidentifies a pedestrian as a fallen branch versus misidentifying the person as a biker. Biker is semantically closer to pedestrian than branch, so the car is correspondingly more likely to take an appropriate action. This metric can also be considered as a "semantic robustness" metric; models which make semantically aligned mistakes under perturbation are more semantically robust than models which make random mistakes.

We propose a method using adversarial training [Goodfellow et al., 2015] to incorporate semantic knowledge into the training process, with the aim of increasing semantic alignment between the model's mistakes and the respective true labels. While there exist other methods of embedding class hierarchical information in neural networks, our approach has the added benefit of offering insight on the relationship of robust and non-robust features [Ilyas et al., 2019] to semantic alignment.

We also evaluate the mistake severity of our method under both adversarial and naturally corrupted conditions – for example, blurring or variations in brightness and saturation [Hendrycks and Dietterich, 2019]. These types of degradations reduce a model's discrimination capability and cause it to make more mistakes. In our study, we use these conditions as a proxy for distribution shift and other sources of error, and aim to decrease mistake severity under these conditions.

The contributions of this work are as follows:

- We propose a method of increasing alignment of semantically similar classes based on targeted adversarial training.

- We show that our approach yields a model that performs better in terms of mistake severity than models trained with standard and common adversarial objectives in multiple degraded conditions.

- We discuss the surprising role of non-robust features in supporting semantic alignment.

## 2  Related Work

Since the notion of adversarial examples was introduced [Szegedy et al., 2014], there have been many works attempting to understand adversarial attacks including [Engstrom et al., 2019, Santurkar et al., 2019, Ilyas et al., 2019]. Ilyas et al. [2019] attributes the presence of adversarial examples to "non-robust" features in the dataset that provide the model with useful signal for classification but are not meaningful (and are often imperceptible) to humans. An adversarially robust network is limited to using "robust" features – features that remain useful for classification even when adversarial perturbations are applied. In other words, Ilyas et al. [2019] posits that there can exist signal in the dataset that is useful for standard classification tasks but can be easily exploited in an adversarial setting.

Current adversarial training techniques are highly focused on pixel-level perturbations, usually in the same flavor as the current "gold standard" robust optimization technique introduced in Madry et al. [2018]. There has been some work to extend this to variations in noise, lighting, or other deviations [Wang et al., 2021a, Saikia et al., 2021]. Namely, there have also been a few examples in the literature around generating semantic adversarial examples which, unlike the classic imperceptible pixel perturbations, are focused on creating inputs that modify semantically meaningful attributes, resulting in images that are still visually faithful, in a semantic sense, to the true label [Wang et al., 2021b, Qiu et al., 2020, Hosseini and Poovendran, 2018].

There also exists prior work that addresses the issue of mistake severity in neural networks, although Bertinetto et al. [2020] notes that despite top-1 accuracies of state-of-the-art classifiers showing steady improvement in performance over the last five years, mistake severity has been stagnant. Bertinetto et al. [2020] also provides a survey of work on "making better mistakes," identifying three main approaches. The first approach they discuss is embedding semantic knowledge in labels, which seeks to modify class representations into an embedding that is more semantically-aligned, for example, by drawing from text sources like Wikipedia. The second approach is using hierarchical losses, i.e., altering the loss function to penalize predictions further away from the true label on a taxonomy tree. The last approach discussed is using hierarchical architectures, incorporating semantic class hierarchy into the classifier without modifying the loss function. They include two of their own variations on standard cross-entropy loss to incorporate prior knowledge into the model.

Within the setting of adversarial training, Ma et al. [2021] introduces the concept of hierarchical adversarial robustness to address mistake severity. Hierarchical adversarial robustness relies on the notion of hierarchical adversarial examples – adversarial examples that cause a misclassification at the "coarse" level (i.e., a misclassification outside of a true label's superclass). Ma et al. [2021] creates a hierarchical network that consists of one network to identify the image's coarse class, and then uses a coarse class-specific network to identify the image's "fine" class. However, the focus of Ma et al. [2021] is on protecting against adversarial examples, whereas we aim to increase the semantic alignment of the model's predictions in order to reduce mistake severity in both adversarial and natural conditions.

## 3   Method

We first define the standard classification task as follows: Let $D$ be the distribution of our data, from which we have input pairs $(x, y)$, where $x \in \mathbb{R}^d$ is a sample point with true label $y$. Given a machine learning model $f_\theta$, parameterized by $\theta$, a loss function can be written as $\mathcal{L}(f_\theta(x), y)$. Then the standard training process has the objective:

$$min_\theta \, \mathbb{E}_{(x,y) \sim D}[\mathcal{L}(f_\theta(x), y)].$$

Additionally, we refer to adversarially robust models as models trained using *untargeted* adversarial training (i.e., finding perturbations that will cause any misclassification, without regards to what the incorrect label is), with the adversarial perturbation specified as $\delta \in \mathbb{R}^d$ and constrained by $\epsilon$. The adversarial training objective is defined as:

$$min_\theta \, max_{\delta:||\delta||<\epsilon} \, \mathbb{E}_{(x,y) \sim D}[\mathcal{L}(f_\theta(x + \delta), y)].$$

We embed semantic knowledge into the training process by using semantically *targeted* adversarial attacks. Unlike the untargeted approach, this approach yields perturbations that will fool the model into predicting a specified (target) class. We ultimately use a staged training approach such that the first stage applies semantically targeted adversarial training and the second stage applies standard training. Our semantically targeted training method is adapted from the targeted version of the adversarial training objective (thus converting formulation of the problem from a min-max to a bi-level optimization approach), with the target $t$ being chosen from a set of semantically similar classes $C(y)$ to the original label $y$. Specifically, our objective function is:

$$min_\theta \, \mathbb{E}_{(x,y) \sim D}[\mathcal{L}(f_\theta(x + \delta^*), y)],$$

where $\delta^* = \underset{||\delta||<\epsilon}{\text{argmin}}[\mathcal{L}(f_\theta(x + \delta), t)]$ is the value needed to achieve the perturbation that causes a misclassification into the target label $t : t \in C(y)$.

## 4   Experiments

We evaluated our method using two notions of semantic similarity: path similarity according to WordNet (i.e., the inverse of the shortest path length between two words in the WordNet structure) [Fellbaum, 1998], and coarse (super) class groupings of labels according to CIFAR100 [Krizhevsky, 2009], which provides 100 fine classes that are grouped into 20 coarse classes.

We let $C(y)$ be a set of five labels with the highest path similarity to $y$. The target label for each adversarial attack was sampled uniformly at random from $C(y)$. All models used a ResNet50 architecture as described in He et al. [2016] and were trained on CIFAR100. We used a learning rate of 0.1, a batch size of 100, and standard values for remaining training parameters. Additionally, data augmentation was applied in the form of random cropping and random horizontal flipping.

To generate the targeted perturbations[1], we used the rAI-toolbox developed by Soklaski et al. [2022] and a 10-step projected gradient descent (PGD) adversary constrained to lie within an $\epsilon$-sized ball in $l_2$ space as proposed by Madry et al. [2018]. The learning rate for the PGD solver was $2.5*\epsilon/10$. We experimented with the value of epsilon across models. Additionally, we experimented with label modification, splitting the label of the perturbed image between the original and target class, to account for large perturbations.

We measured mistake severity as the average path similarity between the model's incorrect predictions and the respective true labels. Additionally, we measured the coarse class accuracy of the model's mistakes (i.e., the proportion of the model's misclassifications that are in the correct coarse class). We found that these two metrics were highly aligned, and trends in the data for the two metrics were almost identical.

The models we compared were as follows:

- **Standard Model**: Model trained with 200 epochs of standard training.

- **Adversarially Robust Model**: Model trained with 200 epochs of untargeted adversarial training with size $\epsilon = 1$ perturbations.

- **Low Epsilon Semantic Targeting Model (LE-SmT)**: Model trained with semantically targeted adversarial training using $\epsilon = 1$ for the $l_2$ perturbation constraint; trained for 200 epochs. The use of a small epsilon in this initial model was based on the standard adversarial framework in which perturbations are imperceptible to the human eye.

- **High Epsilon Semantic Targeting Model (HE-SmT)**: Model trained with semantically targeted adversarial training using $\epsilon = 2.5$ for the $l_2$ perturbation constraint; trained for 200 epochs. We experimented with a higher epsilon in this model to account for the fact that targeted adversarial attacks may be unsuccessful for low values of epsilon, given that the target class may not be near the original class in the embedding space.

- **HE-SmT with Label Modification (HE-SmT-LM)**: Model trained with semantically targeted adversarial training using $\epsilon = 2.5$ for the $l_2$ perturbation constraint; trained for 300 epochs. To better account for large perturbations, we set the labels of the perturbed instances to be a hybrid of the target and original image. We modified the one-hot encoded labels such that the index for the true class and the index for the target class were both set to 0.5.

- **Staged Training Model (ST)**: Model trained with a staged training approach to combine our semantically targeted method with standard training. The motivation behind this method was to apply our semantic targeting method to increase alignment between similar classes, while also leveraging the non-robust signal that seemed to contribute to the baseline standard model's performance. This model was trained with 200 epochs of semantic training (training was set up as specified for HE-SmT-LM) and then an additional 100 epochs of standard training.

We conducted all experiments on a cluster with NVIDIA Tesla V100 GPUs. One epoch of standard training took approximately 1 minute and one epoch of semantically targeted training and untargeted adversarial training took approximately 10 minutes.

## 5 Results

In this section, we will show results on mistake severity against both adversarial perturbations and natural corruptions. We provide evidence that the ST model outperforms the others with this metric.

---

[1]We used the workflow provided by the rAI-toolbox library (subject to the MIT License and FAR 52.227-11 - Patent Rights - Ownership by the Contractor, May 2014) for experiment set-up: `https://github.com/mit-ll-responsible-ai/responsible-ai-toolbox/`

## 5.1 Adversarial Perturbations

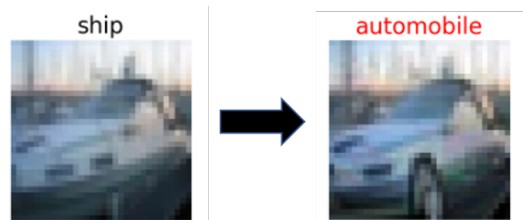

Figure 1: Example of a large perturbation instance. Here, the large perturbation adds a visually perceptible wheel to the ship, causing the model to understandably classify the perturbed instance as an automobile

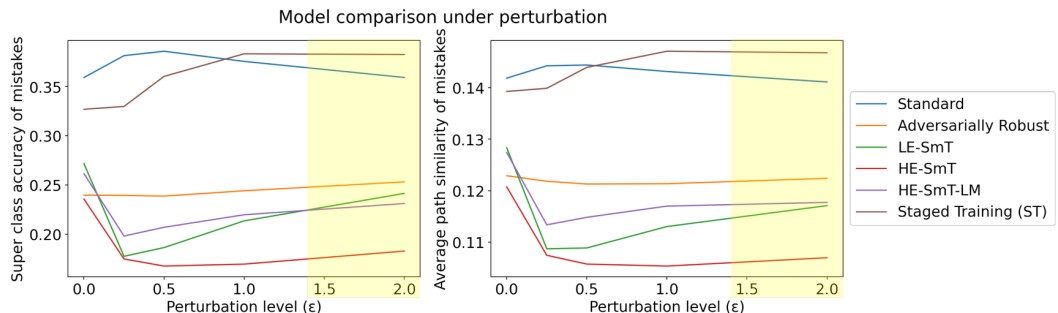

Figure 2: Model comparison of mistake severity over increasingly perturbed data; the plot on the left shows the proportion of misclassified instances where the prediction is in the correct super class, and the plot on the right shows average path similarity between the model's misclassified instances and the respective true labels. The yellow boxes indicate the "high perturbation" region on each plot.

We measured mistake severity over increasingly perturbed data using untargeted perturbations, and especially considered the range of $\epsilon = 1.5$ to $\epsilon = 2.0$ to act as a proxy for data in imperfect conditions, as in our original goal of robustness in challenging conditions. Figure 1 shows an example of a perturbation in this range. Our baseline models were the standard and adversarially robust models. The standard model achieved the best semantic alignment of mistakes on clean data among the compared models. The adversarially robust model had considerably worse mistake severity than the standard model, even on data with high levels of perturbation.

Both the LE-SmT and HE-SmT models failed to improve on mistake severity. We found that the success rate of low epsilon semantically targeted attacks at early stages of training was low compared to untargeted attacks; thus, our other models (HE-SmT-LM and ST) focused on higher values of epsilon to ensure higher success rate of attacks on initially unaligned classes. HE-SmT-LM produced slightly improved results but still failed to compete with the standard model. The ST model recovered some semantic alignment on clean data and also performed better than all other models on highly perturbed data. Results for all models are shown in Fig. 2.

## 5.2 Natural Corruptions

Additionally, we compared the mistake severity of the standard, adversarially robust, and ST models on the CIFAR-100-C dataset [Hendrycks and Dietterich, 2019]. The CIFAR-100-C dataset applies common corruptions such as changes in contrast or blurring to images from CIFAR-100. This dataset allowed us to test model performance under naturally degraded conditions, as it provides natural corruptions at various levels of severity. Corruptions are measured on a scale of 1-5, with a value of 1 being the least severe and 5 being the most severe; we refer to a severity of 1 as low severity and a severity of 5 as high severity. We evaluated model performance on test data for 19 sources of corruption.

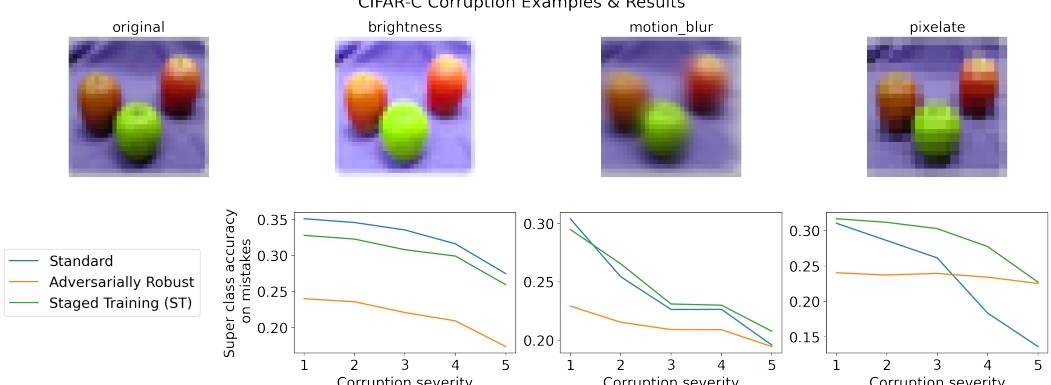

Figure 3: The top row displays instances of different corruption types at a severity of 5. The bottom row compares the super class accuracy on mistakes on the corresponding type of corruption for standard, adversarially robust, and ST models. The corruption types were selected to display different patterns in model performance – in the first, the standard model performs best at all levels of severity. In the second column, the standard model performs best at low severity but ST performs best at high severity. Finally, in the third column, ST performs best at all levels of severity, but the adversarially robust model has competitive performance at the highest severity level.

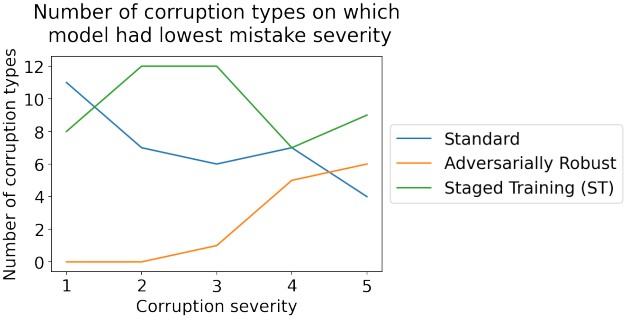

Figure 4: Model comparison of mistake severity over 19 types of common corruptions of increasing severity. The plot measures mistake severity as super class accuracy on mistakes. The semantically trained model performs best on all but the lowest level of severity.

The standard model outperformed both the adversarially robust and ST models on corruptions of low severity, achieving the highest semantic alignment of mistakes on 11/19 corruption types. ST performed best on the remaining 8 corruption types for low severity. However, for corruptions of high severity, ST had the highest semantic alignment of mistakes on 9/19 corruption types The standard model performed best on only 4/19, and the adversarially robust model performed best on the remaining 6/19. Under high severity corruptions, ST outperformed the standard model in terms of mistake severity on 15/19 corruption types. Examples for specific corruptions can be seen in Fig. 3. Results for all levels of severity can be seen in Fig. 4.

These results provide further evidence that ST retains semantic alignment of mistakes for highly degraded conditions, relative to standard and adversarially trained models.

## 6   Discussion and Conclusions

First, we note that many definitions of robustness use accuracy against a specific type of corruption as the success metric; these definitions fail to account for the severity of the model's mistakes. Our experiments demonstrate that there is a gap between current popular robustness techniques and their efficacy when measured against mistake severity. We demonstrate an approach that reduces mistake severity in both adversarially and naturally degraded conditions, indicating that semantic consistency

in mistakes can be improved by incorporating this goal into model training procedures. We hope that these results will motivate further improvements to training procedures with regards to reducing mistake severity.

Second, we observe the role of non-robust signal in semantic alignment of classes. Since non-robust signals are visually imperceptible and don't influence human perception of the semantics of an image, it is intuitive to assume that non-robust signals are less semantically aligned than robust signals, and therefore that robust models should be more semantically aligned. However, we found the opposite in our experiments. Models trained to utilize robust signal performed worse on semantic alignment of mistakes than models without adversarial training. This was true even for semantically targeted adversarial training. Our semantic training approach was only effective in our experiments when the model was allowed to leverage some amount of non-robust signal through staged training. We hypothesize that our two-step training method creates semantic alignment among non-robust features; the ability to use non-robust features when robust features have degraded would then contribute to the increased semantic alignment of mistakes. Alternatively, non-robust features may naturally be more closely related to semantic alignment than originally expected.

Finally, we highlight that semantic alignment does not always reconcile with visual alignment. Since neural networks rely on visual features of the data, it is interesting to consider the extent to which external sources of semantic knowledge can compensate for visual differences in classes. Bertinetto et al. [2020] discusses a similar question of the extent to which the semantic knowledge can be arbitrary (thus breaking any correlation between semantic and visual similarity). Bertinetto et al. [2020] finds that the use of arbitrary semantic hierarchies in their methods caused large deterioration in performance in terms of mistake severity. It would be interesting to explore whether these results hold for our method, and whether the type of perturbation used has an effect on the results.

## 7 Future Work

The results from the staged training approach indicate that our semantic targeting method is contributing to some alignment of classes, but that the restricted use of non-robust signal hinders the success of our method. This opens an interesting avenue for exploring the contribution of non-robust features to the alignment of similar classes, especially given that non-robust signal conventionally refers to data that is meaningless according to human perception.

Future work can compare our method to alternative approaches for embedding semantic knowledge, such as modifying the loss function to penalize worse mistakes. Another direction can be to explore alternate perturbation methods that apply more semantically meaningful perturbations, such as attribute-guided perturbations [Gokhale et al., 2021] or perturbations based on spatial transformations [Xiao et al., 2018].

We also note that in this work, we quantify mistake severity by measuring semantic alignment of classes rather than visual alignment; however, there can be instances in which we may prefer to align a class with a visually similar class rather than a semantically similar class. For instance, an autonomous home helper robot that is sent to retrieve cough syrup may be less trusted if it mistakenly retrieves prescription drugs rather than a bottle of apple juice, even if the prescription drugs are semantically more similar to the true object. Thus, future work can consider the problem of balancing the importance of semantic similarity and visual similarity. In particular, it would be also be useful to explore the potential use of non-robust features to add semantic information that is nonaligned with the visual similarity of classes.

## 8 Broader Impact

We believe that this work shows promise in further advancing the work of robust machine learning and building trust with any embodied system utilizing these algorithms. Furthermore, it encourages a different perspective to the usual definitions of robustness, and explores a metric that has remained stagnant over the last several years of advances. More importantly, we expect that finding new ways to leverage human knowledge and provide models with more semantic alignment will be important as the community seeks to move forward from narrow AI. Not only can semantics provide models with better understanding of the environment, but they can also be a way to mitigate known biases in data - for example, can we encode more "fair" associations that are better in line with today's social

expectations? However, as with almost all research in this field, practitioners need to proceed with care, particularly to avoid embedding biased semantic relationships in a model. Fortunately, machine learning scientists and engineers are becoming more cognizant of these issues and we believe that contributing work that will help build more human-aligned, trustworthy, robust models will lead to safer operations in general.

## Acknowledgments and Disclosure of Funding

DISTRIBUTION STATEMENT A. Approved for public release. Distribution is unlimited.

This material is based upon work supported by the Department of the Air Force under Air Force Contract No. FA8702-15-D-0001. Any opinions, findings, conclusions or recommendations expressed in this material are those of the author(s) and do not necessarily reflect the views of the Department of the Air Force.

© 2022 Massachusetts Institute of Technology.

Delivered to the U.S. Government with Unlimited Rights, as defined in DFARS Part 252.227-7013 or 7014 (Feb 2014). Notwithstanding any copyright notice, U.S. Government rights in this work are defined by DFARS 252.227-7013 or DFARS 252.227-7014 as detailed above. Use of this work other than as specifically authorized by the U.S. Government may violate any copyrights that exist in this work.

The authors would like to thank Drs. Sung-Hyun Son and Sanjeev Mohindra for their support and Ms. Olivia Brown, Dr. Makai Mann, and Dr. Rajmonda Caceres for their time and helpful feedback.

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
