# OpenReview forum: "Addressing Mistake Severity in Neural Networks with Semantic Knowledge"
_NeurIPS.cc/2022/Workshop/TEA — TEA_

### Official Review · Reviewer_5AG3 · 2022-10-19
**Limited technical novelty but new use case**

**Rating:** 6
**Confidence:** 3

**Review:**

This paper extended the ordinary adversarial training recipe to the targeted attack-oriented version to increase the semantic similarity between a model’s predictions and actual labels of misclassified instances. Although the approach seems simple, the idea and the use case are interesting. My specific comments are listed below.

1) Strictly speaking, the targeted version of adversarial training is NO longer the min-max optimization problem. This is because the attack objective is not the opposite of the training objective if the target label is present. In contrast, this becomes a bi-level optimization-based adversarial training formulation. See the related work at [Revisiting and Advancing Fast Adversarial Training Through the Lens of Bi-Level Optimization https://arxiv.org/pdf/2112.12376.pdf ]. Please clarify this.

2) Please add more details for the experiment setup. E.g., the training details and evaluation metrics.

3) Feel free to add more examples in Figure 1.

---

### Official Review · Reviewer_gsdZ · 2022-10-19
**The review for "Addressing Mistake Severity in Neural Networks with Semantic Knowledge"**

**Rating:** 4
**Confidence:** 3

**Review:**

This paper proposes a robust training method that fortifies the model's prediction accuracies to the adversarially corrupted inputs. The core idea is to train the model with noisy inputs whose noises maximize the classification error to the given target labels.

Strong point:
- The proposed method is conceptually straightforward.
- It can be used as an off-the-shelf technic for dealing with adversarial attacks.

Weak point:
- The method is too generic: It is challenging to find the novelty of the proposed method. I'm quite sure we can find similar approaches. Having more related research can enhance the credibility of the authors' claim. Otherwise, weakening the claim in L57, "We propose a novel method of increasing alignment of semantically similar classes based on targeted adversarial training." can be more appropriate.
- Weak empirical evidence: As shown in Figure 3,4, the standard model performs better than the proposed method up to a certain degree of noise or corruption.

---

### Official Review · Reviewer_DNXX · 2022-10-20
**This paper proposed to use targeted adversarial attacks to increase the robustness of the model. The topic of the paper is aligned with the theme of the workshop.**

**Rating:** 6
**Confidence:** 3

**Review:**

The paper studies the robustness of machine learning models. This paper proposed to use targeted adversarial attacks to increase the robustness of the model. Specifically, the approach uses the semantic difference between the prediction and the true label as a measure to quantify the concept of “mistake severity”.  This paper is very well-written and easy to understand. The experiment results demonstrated the effectiveness of the proposed approach. The topic of the paper is aligned with the theme of the workshop.

My major concern is the use of “semantic similarity” to quantify “mistake severity”.  I think mistake severity should depend on downstream tasks, i.e., how the downstream components change their behaviors in response to the prediction errors, instead of the similarities between objects from a semantic perspective. For example, two semantically different objects could induce similar robot behaviors, and two semantically similar objects could also induce very different robot behaviors. Using semantic similarity can not fully capture the mistake severity and the design of a semantic similar set may introduce additional human bias in the downstream tasks. In addition, discussion related to training complicity/time could be included.

---

### Decision · Program_Chairs · 2022-10-21

**Decision:**

Accept

**Comment:**

The paper proposed to use the semantic difference between the predictions and labels to quantify the severity of mistakes, which is used to generate targeted adversarial attacks to improve the robustness of the model. The idea is interesting in general, and the proposed methos is straightforward and easy to use. Please try to address the reviewers' concerns, especially the ones from Reviewer gsdZ, in the final version of the paper.